# Turn Waste into Treasure: Spent Substrates of *Auricularia heimuer* Can Be Used as the Substrate for *Lepista sordida* Cultivation

**Chunge Sheng [1], Chunlei Pan [1], Yanfeng Wang [1,\*], Yinpeng Ma [2], Fei Wang [1], Lei Shi [1], Shurong Wang [3], Jinhe Wang [1], Shuqin Liu [3], Peng Zhang [1], Zitong Liu [1], Haiyang Yu [1] and Jing Zhao [1]**

[1] Mudanjiang Branch, Heilongjiang Academy of Agricultural Sciences, Mudanjiang 157000, China; shengchunge@sina.cn (C.S.); 101pcl@163.com (C.P.); wangfei90wf@163.com (F.W.); shilei_com007@163.com (L.S.); jinhe-happy@163.com (J.W.); zp4479@163.com (P.Z.); lztliuzitong@163.com (Z.L.); haiyang234@126.com (H.Y.); zhaojing8566@126.com (J.Z.)

[2] Institute of Microbiology, Heilongjiang Academy of Sciences, Harbin 150010, China; myp19870315@163.com

[3] College of Food Science and Engineering, Shanxi Agricultural University, Jinzhong 030801, China; wzlj2005@163.com (S.W.); lsq20230824@163.com (S.L.)

\* Correspondence: mdjnks@126.com

**Abstract:** Mudanjiang is a major producer of black wood ear (*Auricularia heimuer*) mushrooms in China. It has been estimated that more than 1.5 million tons of spent substrates of *A. heimuer* (SSA) are produced each year. Most of these are discarded or burned and have become an important source of pollution, which urgently merits research to find appropriate uses for them. To explore the feasibility of SSA as a substrate for cultivating *Lepista sordida* mushrooms, experiments were conducted to assess the effects of the addition of 0, 40%, 73%, and 98% SSA on the days required for the mycelia to fully colonize the substrate and initiate primordia, biological efficiency (BE), yield, and composition of the chemical biomass of the *L. sordida* fruiting bodies. The yield of fruiting bodies with 0, 40%, 73%, and 98% SSA supplementation for three flushes was $3.90 \pm 0.74$ kg m$^{-2}$, $4.06 \pm 0.77$ kg m$^{-2}$, $4.03 \pm 0.62$ kg m$^{-2}$, and $4.51 \pm 0.65$ kg m$^{-2}$, respectively. The addition of 98% SSA significantly delayed the number of days for the mycelia to fully colonize and form primordia by 6 and 3 d, respectively. This treatment also significantly increased the yield and BE by 15.64% compared with that of the control group. The crude polysaccharide content of $25.64 \pm 0.38$ g 100 g$^{-1}$ was higher in the samples grown on the 98% SSA substrate, which was shown to increase by 78.93% compared with that of the control substrate with 73% corn straw ($14.33 \pm 0.03$ g 100 g$^{-1}$). The content of crude protein of $51.10 \pm 0.08$ g 100 g$^{-1}$ was higher in the samples grown on the 40% SSA substrate, which increased by 11.14% compared with the protein content of the control group. This study reveals that SSA would be a good substrate for the cultivation of *L. sordida* and is an efficient, promising, and cost-effective substrate additive that can improve the quality and yield of these mushrooms, while substantially reducing the problems of disposing of SSA.

**Keywords:** *Lepista sordida*; spent substrates of *Auricularia heimuer* (SSA); grass-rotting fungus

## 1. Introduction

Wild *Lepista sordida* resources are widely distributed over Asia, North America, and Europe [1]. In China, *L. sordida* is mainly distributed in the northeastern, northern, central, and southern parts of the country [2]. These mushrooms are famous for their flavor and the high nutritional value of their protein [3], beneficial mineral elements [4], and amino acids (including eight essential amino acids) [5] among others, and are popular with consumers and researchers. Recently, polysaccharides [6], chlorinated sesquiterpenes [7], and other bioactive compounds from *L. sordida* were extensively studied and found to have anti-cancer [8], immune regulatory [9], anti-aging [10], antioxidative, and hepatoprotective [11]

properties under both in vivo and in vitro conditions from both submerged cultures and fruiting bodies.

As a functional edible fungus, the species *L. sordida* is appreciated globally. However, its cultivation is still largely unexplored. As early as the 1980s, mushroom researchers in China began to explore the artificial cultivation technology of this mushroom, and some progress has been made over the previous decade. Tian et al. [12] cultivated the fungi by fermentation and clinker cultivation with straw. Their study explained 19.9% of the highest biological efficiency (BE); that of clinker cultivation was higher than that of cultivation by fermentation. Lun [13] proposed clinker cultivation with corn straw and explained 36.8% of the BE on a Kentucky bluegrass (*Poa pratensis*) lawn by bionics wild cultivation. In addition, clinker cultivation with corncobs was also conducted and resulted in the highest BE of 28.4%. Wild strains of *L. sordida* have been domesticated previously in China with rice straw composts and produced the greatest yield of 2.6 kg pot$^{-1}$ using 15 kg of compost, while Li et al. [14] obtained a BE of 43%. Sawdust and lawn grass composts were used as substrates by Zhou et al. [15], which explained 21.1 g bag$^{-1}$. A study by Xu et al. [16] utilized corncob composts and confirmed 41.22% of the BE. Successful cultivation of *L. sordida* was reported by Tongbai from Thailand in 2017 [1]. In their research, the wild strain of *L. sordida* was cultivated on the compost substrate with rice straw as the main substrate, with a yield of 93–287.5 g kg$^{-1}$. We can conclude from the studies conducted to date that the methods of cultivating *L. sordida* mainly include clinker bag mulch and fermented bed mulch. The main raw materials for cultivation are sawdust, rice (*Oryza sativa*) straw, wheat (*Triticum aestivum*) straw, corn (*Zea mays*) straw, corncob, cotton (*Gossypium hirsutum*) seed hull, and other types of agricultural wastes. *L. sordida* has not yet been cultivated commercially because of the instability of the strain and the limitation of culture media and conditions.

Mudanjiang City is located in Northeast China. As "the capital of wood ear in the world", it is the main area of production of black wood ears in China and even the world. As estimated, more than three billion bags of black wood ears were cultivated in Mudanjiang City in 2021 (China Edible Fungi Association). By calculating the substrate as 0.6 kg bag$^{-1}$ of mushrooms, 1.8 million tons of spent substrates of *Auricularia heimuer* [17] (SSA) have been produced. However, there are no suitable ways to dispose of this residue. Most of it has been abandoned or burned for energy, which is neither environmentally friendly nor economical. Therefore, how to effectively solve the problem of comprehensive utilization of SSA has become a hot and urgent research topic. In recent years, numerous studies on the re-utilization of spent mushroom substrate (SMS) to cultivate the same or different species of mushroom have been conducted [18,19]. However, there are few studies on the cultivation of mushrooms using SSA composts, and they have only been reported from China. Such studies include the cultivation of shaggy mane (*Coprinus comatus*) [20], *Agaricus bisporus* [21], and *Hohenbuehelia serotina* [22] using SSA. However, to our knowledge, there have been no studies on the re-utilization of SSA to cultivate *L. sordida*.

The price of dried mushrooms ranges from CNY 200 to CNY 600 per kilogram, and producing them economically was the objective. This study explored the effects of SSA on the mushrooms produced by *L. sordida* by analyzing their yield, growth, and nutritional composition. This not only provides data to support the development of new substrates to cultivate *L. sordida* but also provides a new way to reutilize the SSA resources.

## 2. Materials and Methods

### 2.1. Lepista Sordida Strain

The *L. sordida* strain ZD3 used in this study was isolated from a wild strain that was collected in the suburbs of Mudanjiang City. It was preserved in the Mudanjiang Branch of Heilongjiang Academy of Agricultural Sciences (Mudanjiang, China), and a partial sequence has been deposited in GenBank as accession number MZ298493. The pure culture was inoculated on potato dextrose agar (PDA; 200 g L$^{-1}$ diced potatoes, 20 g L$^{-1}$ glucose, and 15 g L$^{-1}$ agar) medium at 25 °C.

*2.2. Substrate Preparation*

2.2.1. Preparation of SSA and Corn Straw

SSA Preparation

The SSA used in this study was provided by the Mudanjiang Branch of Heilongjiang Academy of Agricultural Sciences (Mudanjiang, China). Uncontaminated SSA was selected. The plastic bag was peeled off, and the SSA was then crushed with a grinder (Xinzhuohui-FSJ, Zhuohui Machinery Co., Ltd., Zhengzhou, China) to pass through an inner screen with 2 cm holes.

Corn Straw Preparation

The corn straw used in this study was provided by the Mudanjiang Branch of Heilongjiang Academy of Agricultural Sciences (Mudanjiang, China). The corn straw branches were cut into small pieces of 5–10 cm using a crusher (M-ZR, Nongxing Machinery and Equipment Factory, Xingyang, China). All the materials used in this study were dried in the sun.

2.2.2. Analysis of the Proximate Components of SSA

The dry matter (DM) of the samples was determined after they had been dried to a constant weight at 65 °C. The contents of total carbon (TC) and total nitrogen (TN) were estimated by the loss of ignition and Kjeldahl methods, respectively. The C/N ratio is the ratio of the TC in the substrate to that of the TN. The contents of ash-free neutral detergent fiber (NDF) [23], ash-free acid detergent fiber (ADF), and lignin [24] were determined as described. The hemicellulose content was calculated as the difference between NDF and ADF and that of cellulose as the difference between ADF and acid detergent lignin (ADL).

*2.3. Substrate Formula*

Four different formulas were used to cultivate *L. sordida*. The composition of the substrates used in this study is shown in Table 1.

**Table 1.** Composition of substrate ingredients for the cultivation of *Lepista sordida* (dry matter) (*w/w*).

| Material | Treatment Group | | | |
|---|---|---|---|---|
| | **CK [25]** | **T1** | **T2** | **T3** |
| Corn straw | 73 | 33 | 0 | 0 |
| SSA | 0 | 40 | 73 | 98 |
| Cow dung | 25 | 25 | 25 | 0 |
| Gypsum | 1 | 1 | 1 | 1 |
| Lime | 1 | 1 | 1 | 1 |

Note: CK: control; SSA: spent substrates of *Auricularia heimuer*.

*2.4. The Specific Manner of Fermentation*

The material was prepared as described by the proportion of formula. The corn straw was pre-wetted, and the material was mixed evenly. The mixture was placed in a trapezoidal pile with two rows of air holes that had a diameter of 5 cm and were spaced at 50 cm intervals. They were punched vertically on the material pile. When the temperature was above 60 °C (20 cm below the top of the pile), the pile was maintained for more than 24 h. At this time, the pile was re-mixed and re-piled into a trapezoid. The purpose of this step was to discharge harmful gases and add more fresh air so that the fermentation process would be more successful. In general, four times of re-piling are needed during the whole fermentation period. After the first re-piling, the interval was 6, 5, and 4 d before the next re-piling, and the whole fermentation process should last 20~25 d. The fermented medium was free of ammonia and fecal matter, elastic, and non-lumpy and non-sticky. Finally, 1% gypsum was mixed with the substrate, and its pH was adjusted to 7.0~7.5 using lime. The contents of water in the final mixtures were adjusted to 55–65% (*w/w*).

## 2.5. Key Points of Cultivation

The ridge cultivation model was used in the greenhouse, and the prepared substrates were placed in each cultivation area (15 m$^2$) at a density of 13 kg m$^{-2}$ (dry weight). A volume of 10% (*w/w*) (dry weight) of the spawn was spread on the fermented substrates for inoculation. There were three replicates of the ridges per treatment that were randomly distributed in the greenhouses. At the time that the mycelia colonized the substrate, the greenhouse was maintained at 20–26 °C and 70−75% relative humidity (RH). The greenhouse was kept at 22–26 °C and 80–90% RH during the times of the initiation of primordia and the growth of fruiting bodies.

## 2.6. Investigation of Agronomic Traits

The total yield of three flushes of mushrooms was determined in a harvest period. Three random replicates were used. The yield was calculated as shown in Equation (1):

$$\text{Yield (kg m}^{-2}) = \Sigma \text{ weight of 3 flushes freshly harvested mushrooms}/15 \qquad (1)$$

The biological efficiency (BE) was calculated as shown in Equation (2):

$$\text{BE (\%)} = (\text{weight of freshly harvested mushrooms}/\text{weight of substrate dry matter}) \times 100\% \qquad (2)$$

The growth parameters included the diameter of pileus (cm) and its thickness (cm). The length and diameter of the stipe were measured in cm from 30 random replicates (fruiting bodies) in the first flush of mushrooms (10 fruiting bodies in each replicated ridge). The parameters that were measured and noted from each replicate included the time of colonization by the mycelia. This represented the days that it took for the substrate to be fully colonized by the inoculated mycelium. In addition, the time of primordial initiation was also measured based on the number of days that it took the mycelium to grow from covering the soil to producing pinhead fruiting bodies.

## 2.7. Chemical Analysis

Fruiting bodies of *L. sordida* were randomly collected from each treatment group. The amount was in equal proportion to the first flush. The samples were dried to a constant weight at 60 °C in an oven and then stored at 4 °C. The chemical analyses were conducted by the Heilongjiang Huace Testing International Corporation (Harbin, China). The contents of crude protein, crude fat, ash, crude polysaccharide, and crude fiber were determined as described by Zou et al. [26].

## 2.8. Statistical Analysis

The original data were processed using Microsoft Excel 2019 (Redmond, WA, USA). LSD multiple range tests were used to assess the differences among the means of four treatments at a 95% confidence level ($p < 0.05$) as shown by a one-way analysis of variance (ANOVA). The statistical analyses were conducted using SPSS 26.0 for Windows (IBM, Inc., Armonk, NY, USA). The data obtained were expressed as the means $\pm$ SD (standard deviation). The figures were plotted using GraphPad Prism 8 (San Diego, CA, USA) and modified manually.

## 3. Results

### 3.1. Components of SSA

The TC content of the SSA was 42.18%, while that of the TN was 1.89%. The C/N ratio of the SSA was 22:1. The contents of cellulose, lignin, and hemicellulose in the SSA were 16.00%, 14.90%, and 39.10%, respectively.

### 3.2. Mycelia Growth and Primordial Initiation

The mycelium colonization time and primordial initiation time of *L. sordida* cultivated on different treatment substrates are shown in Figure 1 and supplementary Table S1. It

took 19.33 ± 0.58 d for the mycelia to colonize the substrate in the CK treatment with 73% corn straw. The values for the T1 treatment with 40% SSA replacement corn straw, the T2 treatment with 73% SSA replacement corn straw, and the T3 treatment with pure SSA were 21.67 ± 0.58 d, 23.67 ± 0.58 d, and 25.67 ± 0.58 d, respectively. There was a significant difference among each treatment group. As the proportion of SSA increased, the days of mycelial colonization time increased. The time that it took for the mycelia to colonize the substrate in the T3 treatment with pure SSA was the longest, which was delayed by 6 d compared with that of the CK treatment with 73% corn stalk.

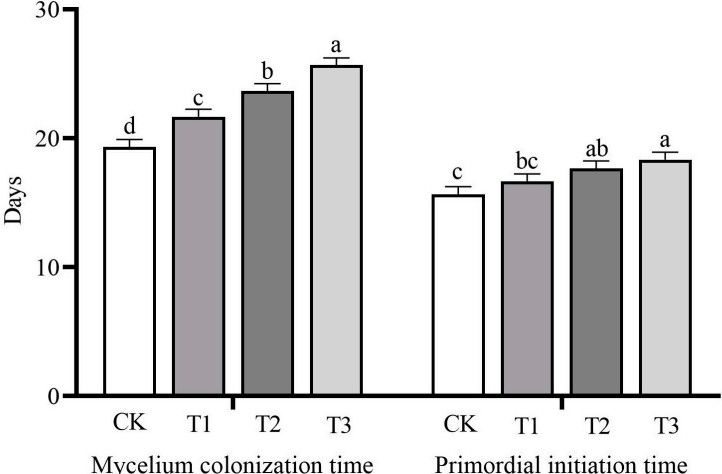

**Figure 1.** Key point of hyphal development. The means ± SD are shown. Different lowercase letters above each bar in a given group indicate significant differences ($\alpha$ = 0.05, ANOVA, LSD test). CK: treatment with 73% corn straw, 25% cow dung, 1% gypsum, and 1% lime; T1: treatment with 33% corn straw, 40% SSA, 25% cow dung, 1% gypsum, and 1% lime; T2: treatment with 73% SSA, 25% cow dung, 1% gypsum, and 1% lime; T3: treatment with 98% SSA, 1% gypsum, and 1% lime. ANOVA, one-way analysis of variance; CK, control; SD, standard deviation; SSA, spent substrate of *Auricularia heimuer*.

The number of days required for primordial initiation on the CK group with 73% corn straw was 15.67 ± 0.58. Those of the T1 treatment with 40% SSA replacement corn straw, the T2 treatment with 73% SSA replacement corn straw, and the T3 treatment with pure SSA were 16.67 ± 0.58 d, 17.67 ± 0.58 d, and 18.33 ± 0.58 d, respectively. The days of primordial initiation time in the T1 group had no significant difference than that of the CK group. When the SSA was replaced with more than 73% corn straw, the days required for the time of primordial appearance were delayed significantly. The T3 group was delayed by 3 d compared with that in the CK group.

### 3.3. Fruiting Body Morphology

As shown in Figure 2 and supplementary Table S1, the SSA supplements differentially affected the diameters of the cap and stipe, cap thickness, and stipe length. For the cap diameter, the replacement of SSA with 40% corn straw had no significant effect on the cap diameter, but when the replacement ratio was more than 73%, the cap diameter decreased significantly. The effect of SSA supplementation on the cap thickness was not significant until 98% of the SSA was replaced, which was significantly higher than those of the CK, T1, and T2 groups. The replacement of SSA with 40% of corn straw had no significant effect on the stipe thickness. However, when the replacement ratio was more than 73%, the stipe thickness increased significantly. The T3 group with 98% SSA supplement gave the largest stipe thickness, which indicated that the stipe thickness increased with the increase in the SSA replacement ratio. However, the stipe lengths of the fungi grown on each treatment were not significantly different, which showed that the SSA addition and supplement ratio had no effect on the stipe length of *L. sordida*.

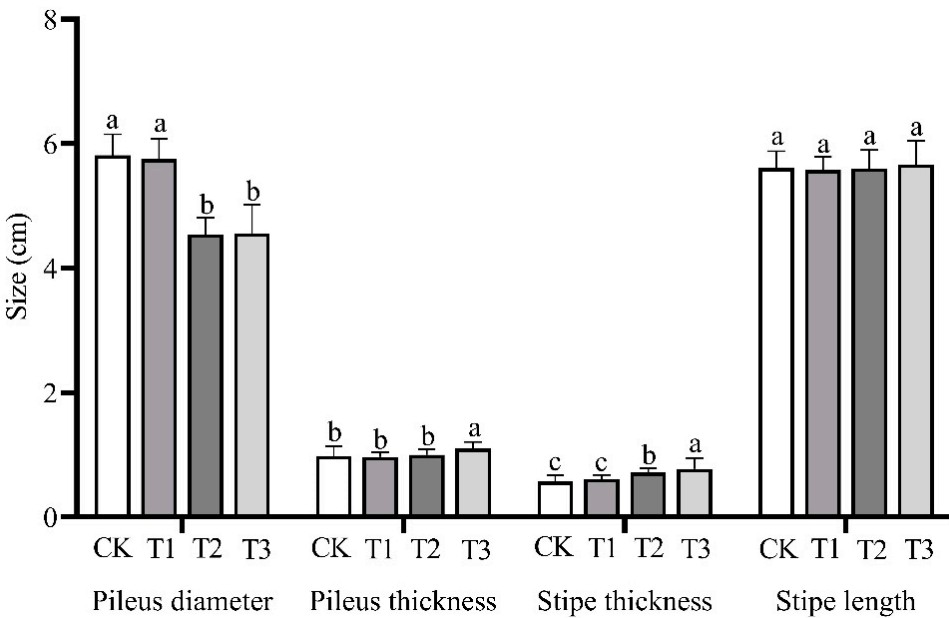

**Figure 2.** Morphological characteristics of the *Lepista sordida* fruiting bodies grown on different substrates. The means ± SD are shown. Different lowercase letters above each bar in a given group indicate significant differences ($\alpha$ = 0.05, ANOVA, LSD test). CK: treatment with 73% corn straw, 25% cow dung, 1% gypsum, and 1% lime; T1: treatment with 33% corn straw, 40% SSA, 25% cow dung, 1% gypsum, and 1% lime; T2: treatment with 73% SSA, 25% cow dung, 1% gypsum, and 1% lime; and T3: treatment with 98% SSA, 1% gypsum, and 1% lime. ANOVA, one-way analysis of variance; CK, control; SD, standard deviation; SSA, spent substrates of *Auricularia heimuer*.

### 3.4. Yield and Biological Efficiency

Cultivation continued for 75–90 d during which three flushes were harvested. As shown in Table 2 and Figure 3, the total fresh weights of the mushrooms grown on the CK, T1, and T2 treatments were 3.90 ± 0.74 kg m$^{-2}$, 4.06 ± 0.77 kg m$^{-2}$, and 4.03 ± 0.62 kg m$^{-2}$, respectively. The total yield among the three groups did not differ significantly. The T3 group produced the highest total yield of 4.51 ± 0.65 kg m$^{-2}$, which was significantly higher than that of the other treatments. The first flush of mushrooms contributed approximately 50% to the total yield, and the second flush contributed 30% to the total yield.

**Table 2.** Comparison of the biological efficiency (BE) and fresh weight of *Lepista sordida* mushrooms grown on different treatment groups (means ± SD).

| Treatment Group | Fresh Weight of the Mushrooms by Flushes (kg m$^{-2}$) | | | | BE (%) |
| :---: | :---: | :---: | :---: | :---: | :---: |
| | First Flush | Second Flush | Third Flush | Total Fresh Weight (kg m$^{-2}$) | |
| CK | 2.09 ± 0.02 ab | 1.17 ± 0.13 a | 0.64 ± 0.13 c | 3.90 ± 0.74 b | 30.00 ± 0.77 b |
| T1 | 2.18 ± 0.03 ab | 1.22 ± 0.05 a | 0.66 ± 0.09 bc | 4.06 ± 0.77 b | 31.18 ± 0.62 b |
| T2 | 2.05 ± 0.16 b | 1.10 ± 0.07 a | 0.88 ± 0.11 ab | 4.03 ± 0.62 b | 30.97 ± 1.79 b |
| T3 | 2.24 ± 0.09 a | 1.24 ± 0.10 a | 1.03 ± 0.15 a | 4.51 ± 0.65 a | 34.69 ± 1.03 a |

Note: The same letter indicates no significant differences at $p < 0.05$ according to LSD multiple range tests. Different lowercase letters indicate significant differences in each column ($p < 0.05$). CK: treatment with 73% corn straw, 25% cow dung, 1% gypsum, and 1% lime; T1: treatment with 33% corn straw, 40% SSA, 25% cow dung, 1% gypsum, and 1% lime; T2: treatment with 73% SSA, 25% cow dung, 1% gypsum, and 1% lime; and T3: treatment with 98% SSA, 1% gypsum, and 1% lime. CK, control; SSA, spent substrates of *Auricularia heimuer*.

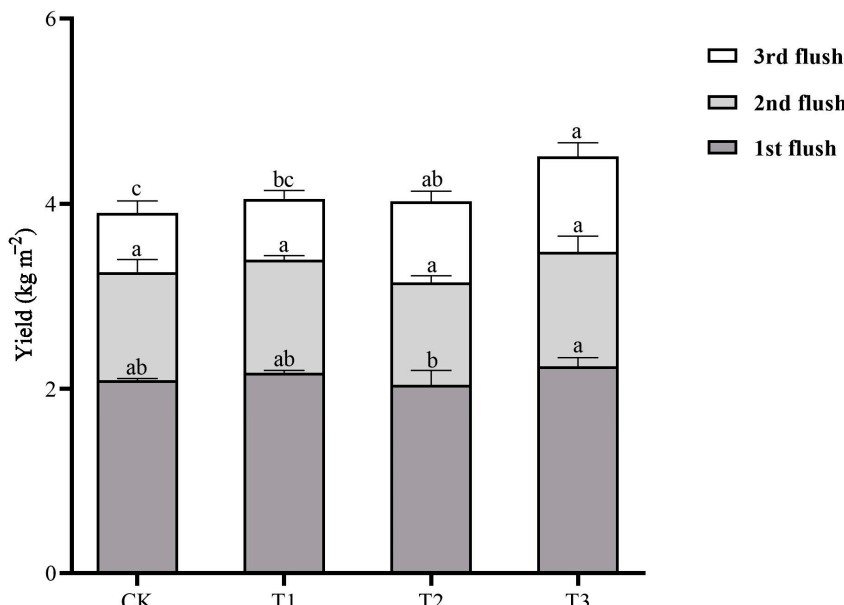

**Figure 3.** Column map of yield distribution. The yield distributions for the third, second, and first flushes are shown from top to bottom, respectively. The means ± SD are presented. Different lowercase letters above each bar in each flush indicate significant differences ($\alpha$ = 0.05, ANOVA, LSD test). CK: treatment with 73% corn straw, 25% cow dung, 1% gypsum, and 1% lime; T1: treatment with 33% corn straw, 40% SSA, 25% cow dung, 1% gypsum, and 1% lime; T2: treatment with 73% SSA, 25% cow dung, 1% gypsum, and 1% lime; and T3: treatment with 98% SSA, 1% gypsum, and 1% lime. ANOVA, one-way analysis of variance; CK, control; SD, standard deviation; SSA, spent substrates of *Auricularia heimuer*.

The fresh weight of the first flush mushrooms in the different treatments ranged from 2.05 ± 0.16 kg m$^{-2}$ to 2.24 ± 0.09 kg m$^{-2}$. The fresh weight in the first flush of mushrooms did not differ significantly among the CK, T1, and T3 groups except for the T2 group, which had a significantly lower yield than that of the other groups. The fresh weight of the second flush of mushrooms ranged from 1.10 ± 0.07 kg m$^{-2}$ to 1.24 ± 0.10 kg m$^{-2}$. However, the treatments did not differ significantly. The effect of the SSA substrate on the yield of the third flush mushrooms was significant. The fresh weight of mushrooms grown on the T1 group was not significantly different than that of the CK group. When the SSA was replaced with more than 73% corn straw, the fresh weight of mushrooms increased significantly.

The BE of *L. sordida* mushrooms grown on the CK, T1, and T2 groups were 30.00 ± 0.77%, 31.18 ± 0.62%, and 30.97 ± 1.79%, respectively. There was no significant difference among each treatment. The T3 group produced the highest BE of 34.69 ± 1.03%, which was significantly higher than that of the other treatments.

### 3.5. Nutritional Value

The nutritional value of *L. sordida* cultivated on different substrates were shown in Figure 4 and supplementary Table S2. The contents of crude fiber, crude protein, and ash of the fruiting bodies grown on the TI treatment with 40% SSA replacement corn straw were significantly higher than those of the other treatments, while the contents of crude fat and crude polysaccharide were significantly lower than those of the other treatments. Compared with the CK group, the contents of crude fiber, crude polysaccharide, and ash in the samples grown on the T2 group decreased significantly, while the contents of crude fat and crude protein increased significantly. The samples grown on the T3 treatment had the highest content of crude polysaccharide at 25.64 ± 0.38 g 100 g$^{-1}$ and increased significantly by 78.93% compared with the 14.33 ± 0.03 g 100 g$^{-1}$ of the CK group. In contrast, the contents of crude fat, crude protein, and ash in the samples grown on the

T3 group were significantly lower than those in the other treatments. The content of crude protein of $31.0 \pm 1.08$ g $100$ g$^{-1}$ in the fruiting bodies harvested from the T3 group decreased significantly by 32.61% compared with that of $45.98 \pm 0.08$ g $100$ g$^{-1}$ in the CK group. In addition, in this study, we found that with the increase in SSA supplementation, the accumulation of crude polysaccharides increased, while the accumulation of crude protein and ash decreased.

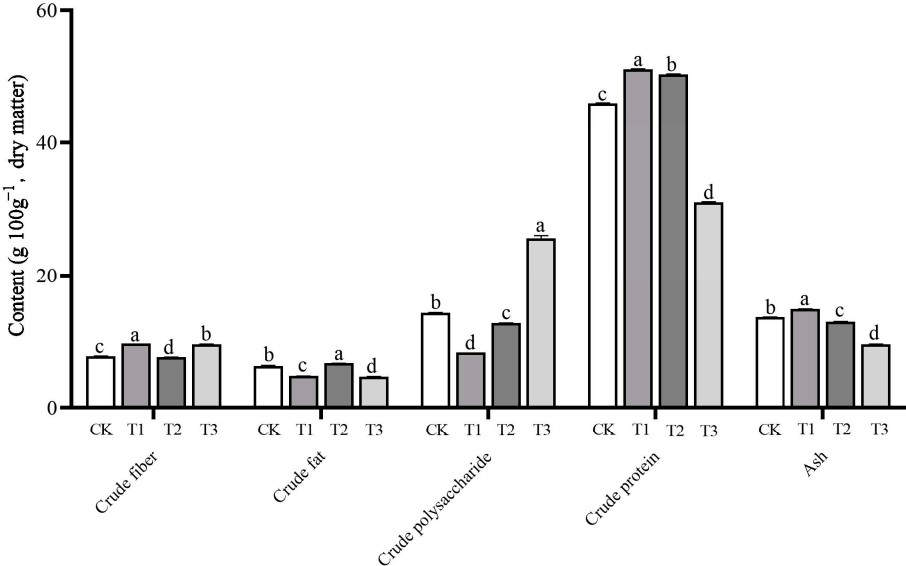

**Figure 4.** Comparison of the chemical composition of *Lepista sordida* on different treatment groups (g $100$ g$^{-1}$ of dry matter). The means $\pm$ SD are shown. Different lowercase letters above each bar in each flush indicate significant differences ($\alpha = 0.05$, ANOVA, LSD test). CK: treatment with 73% corn straw, 25% cow dung, 1% gypsum, and 1% lime; T1: treatment with 33% corn straw, 40% SSA, 25% cow dung, 1% gypsum, and 1% lime; T2: treatment with 73% SSA, 25% cow dung, 1% gypsum, and 1% lime; and T3: treatment with 98% SSA, 1% gypsum, and 1% lime. ANOVA, one-way analysis of variance; CK, control; SD, standard deviation; SSA, spent substrates of *Auricularia heimuer*.

## 4. Discussion

Spent mushroom substrate (SMS) is the lignocellulosic byproduct of the cultivation of mushrooms. It has not been degraded completely because the degradation efficiency of the edible species of fungus only reached 40–80% [27]. Recycling SMS after harvesting the first mushroom cultivation and reusing it in new substrate formulation for mushroom cultivation is a good example of a circular economy. The remaining fibers and nutrients in the mushroom residues can still provide a sufficient source of C for the new cultivation cycles of certain edible mushrooms after they have been treated properly to achieve the goals of saving costs and increasing the overall yield.

The mushroom cultivation model of primary decomposers–secondary decomposers [28] has achieved good results in the following studies, such as the models of *Pleurotus sajor-caju* mushroom–*Agaricus blazei* mushroom pattern [29] and the *Pleurotus eryngii* mushroom–*Volvariella volvacea* mushroom pattern [30]. This study shows that the *Auricularia heimuer* mushroom–*Lepista sordida* mushroom pattern is also an effective innovative mushroom cultivation model. In addition, the yield and crude polysaccharide content of *L. sordida* fruiting bodies grown on 98% SSA were $4.51 \pm 0.65$ kg m$^{-2}$ and $25.64 \pm 0.38$ g $100$ g$^{-1}$, respectively, which were increased by 15.64% and 78.93% than those of the control group. The crude protein content of *L. sordida* fruiting bodies harvested from 40% SSA reached $51.10 \pm 0.08$ g $100$ g$^{-1}$, which increased by 11.14% compared with that of the control group. This confirmed that SSA is a good substrate for cultivating *L. sordida* mushrooms, which can not only increase yield but also improve their quality. As for why SSA affects the content

accumulation of crude protein and crude polysaccharides in the mushrooms, this is a very important issue that requires further research.

In this study, *L. sordida* was successfully cultivated on four treatments. However, the quicker rate of mycelium growth did not correspond with the highest BE and yield. These results were similar to those of studies on *Pleurotus ostreatus* that were cultivated on different byproducts of lignocellulosic material [31] and *Agrocybe cylindracea* cultivated on spent *Pleurotus* compost [32]. In addition, the yield distribution was compared. The distribution of crop yield among the individual flushes was similar on all the substrates. Approximately 50% of the total yield was obtained during the first flush and 30% was in the second flush on all the substrates. The first two flushes significantly contributed to the ultimate yield and BE percentage values (80%). The yields and BE of the first flush were higher than those of the second flush in each treatment. These results were also similar to a study of *A. cylindracea* and *P. ostreatus* that were cultivated on nine types of agro-industrial and forestry byproducts [33] and *P. ostreatus* that was cultivated on different proportions of agro-wastes [34]. However, it was not consistent with the results from the cultivation of *Auricularia polytricha* on the sawdust wastes of spent mushrooms [35], which showed that the BE and yields of the second flush were higher than those of the first flush. These findings were similar to those on the cultivation of *Pleurotus citrinopileatus* using grass plants [36].

In general, the lignin content of SSA was higher than that of corn straw. Although the T3 group had significantly higher total yields than the CK group when grown on crop straw substrate, the yields of the first and second flushes of mushrooms in the T3 treatment were clearly not high. The third flush was significantly higher. The crop straw group did not grow sufficiently. As a kind of grass rot fungus, *L. sordida* showed a higher yield and BE on the SSA substrate with a higher lignin content compared with the traditional crop straw substrate with a lower content of lignin, which was not predicted. We hypothesized that the reason may be that SSA with a higher lignin content, greater density, stronger water retention capacity, and poor dissolved oxygen was a more suitable source of slow-release energy for the growth of *L. sordida* mushrooms. This resulted in a delay in the times of mycelial colonization and initiation of the primordia. With the increase in microbial activity and metabolic products, most of the energy of SSA was utilized after the first two flushes of mushrooms were harvested, which resulted in a high yield of *L. sordida*. The results of this study differ from those of Rizki et al. [37], who found that the relatively higher content of lignin in the substrate inhibited the primordial formation and increased the fruiting body yield, whereas the addition of high concentrations of lignin decreased yield.

The stipe length was not affected by substrate in the present research. This finding was not consistent with that of studies on the cultivation of oyster mushrooms on rice/wheat straw [38], which reported that the stipe length of the fruiting body was affected by the substrate. In this study, after three harvests, there were no more data on yield because high-quality mushrooms were obtained over a short time period as described by Xu et al. [39].

Recently, there have been many studies on the analyses of the chemical contents of mushrooms [40]. They have shown that the chemical contents are easily affected by many factors, including the origins of the substrates, genotypes of the strains, and atmospheric conditions among other factors, which typically differ between studies [41]. According to Lu et al. [3], the wild dried *L. sordida* mushroom contains 43.03 g $100 \text{ g}^{-1}$ of crude protein, 29.70 g $100 \text{ g}^{-1}$ of carbohydrates, 7.58 g $100 \text{ g}^{-1}$ of crude fiber, and 2.01 g $100 \text{ g}^{-1}$ of crude fat. Here, the crude protein contents varied from $31.01 \pm 0.08$ g $100 \text{ g}^{-1}$ to $51.10 \pm 0.08$ g $100 \text{ g}^{-1}$, and the crude fiber contents varied from $7.62 \pm 0.03$ g $100 \text{ g}^{-1}$ to $9.71 \pm 0.01$ g $100 \text{ g}^{-1}$ and were higher than those of Lu et al. [3]. The content of crude fat (varied from $4.70 \pm 0.04$ to $6.71 \pm 0.03$ g $100 \text{ g}^{-1}$) was also higher than that of Lu et al. [3] with 2.01 g $100 \text{ g}^{-1}$. To our knowledge, there have been no studies on the nutritional composition of the other species of mushrooms cultivated on SSA. Krystyna [42] showed that the polysaccharide contents differed depending on the choice of growth substrate. In this study, it is conceivable that supplementing the cultures with a high concentration of SSA

could significantly improve the accumulation of polysaccharose in *L. sordida*, while reducing the accumulation of proteins and ash. In addition, it shows that the SSA substrates can be used to produce high-quality *L. sordida* mushrooms to meet the demands of consumers.

## 5. Conclusions

We examined the effects of SSA substrates on the yield, growth period, BE, nutritional value, and other agronomic characteristics of the edible mushroom *L. sordida*. This study found that the use of the SSA substrate improved the total yield, BE, and nutritional value by influencing the accumulation of fat, protein, ash, polysaccharides, and fiber. SSA is a promising lignocellulosic substrate that can be recycled to cultivate *L. sordida*. Therefore, the SSA substrate has a high potential for the commercial production of *L. sordida* mushrooms. Additionally, the *Auricularia heimuer* mushroom–*L. sordida* mushroom cultivation pattern was established, which greatly improves the efficiency of substrate recycling. However, it still has a long way to go for commercial production. The SSA's highly efficient utilization of cultivated *L. sordida* still merits further research.

**Supplementary Materials:** The following supporting information can be downloaded at: https://www.mdpi.com/article/10.3390/horticulturae9101074/s1. Table S1: Agronomic characteristics of *Lepista sordida* mushrooms grown on different substrates. Table S2: Nutritional value of *Lepista sordida* mushroom grown on different substrates.

**Author Contributions:** Conceptualization, C.S. and Y.W.; methodology, C.P.; software, S.L.; validation, Y.M.; formal analysis, S.W.; investigation, C.S.; resources, H.Y. and Z.L.; data curation, Y.M.; writing—original draft preparation, C.S.; writing—review and editing, J.W. and P.Z.; visualization, L.S.; supervision, J.Z. and F.W.; project administration, C.S.; funding acquisition, Y.W. All authors have read and agreed to the published version of the manuscript.

**Funding:** This research was funded by the Agricultural Science and Technology Innovation Leapfrog Program of Heilongjiang Province, and the Agricultural Characteristic Industry Science and Technology Innovation project (No. CX23TS15).

**Data Availability Statement:** Not applicable.

**Acknowledgments:** We wish to thank Mengran Zhao from the Institute of Agricultural Resources and Regional Planning of the Chinese Academy of Agricultural Sciences for her guidance and selfless assistance in writing and revising this paper.

**Conflicts of Interest:** The authors declare no conflict of interest.

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
