# Peer review of "Turn Waste into Treasure: Spent Substrates of Auricularia heimuer Can Be Used as the Substrate for Lepista sordida Cultivation"

_horticulturae, doi:10.3390/horticulturae9101074_

Round 1
Reviewer 1 Report
The manuscript presents an interesting alternative to use SSA for production of Lepista sordida under circular economy concept.
Neverteless, even quantitative data is provided, only descriptive results (both in tables and figures) and discussion it is not enough for a better comprehension.
It seems that some pictures ang graphical pictures about cultivation systems should improve the paper quality.
I recommend including a graphical abstract and corresponding pictures (for example for data showed in figure 2.
Author Response
Dear Reviewer:
Thank you for your valuable suggestion. We revised the manuscript and added a graphical abstract to improve the quality of this paper. The graphical abstract was reorganized ,please see the attachment.

Reviewer 2 Report
This manuscript is about of use of residue of substrate uded for Auricularia heimuer and the utilization now for the grown of Lepista sordida, I consider that the document required only any corrections that I send in file pdf. Congratulations.

Author Response
Dear Reviewer :
Thank you for your letter and the comments about our manuscript entitled “Turn Waste into Treasure: Spent Substrates of Auricularia heimuer Can be Used as the Substrate for Lepista sordida Cultivation” (horticulturae-2595871). These comments are valuable and will be very helpful for us to revise and improve our paper, as well as to help provide the important guiding significance of our research. We studied the comments carefully and made corrections that we hope will be met with your approval. The revised portions are marked in yellow in the paper.
- Line 108, page 3: “the word is Corn, change!”
We are so sorry for the incorrect description. This is a spelling mistake. “Cron” has been corrected to “Corn.”
- Line 121, page 3: “indicate mean of CK”
Thank you for your valuable suggestion. The CK means control. At the suggestion of another reviewer, we deleted lines 121-125.
- Line 160-161, page 4; line 195, page 5 and line 312, page 9: “mycelium, change word”
Thank you for your valuable suggestions. These words have all been corrected throughout the manuscript.
- Line 195, page 5 ; line 233, page 7 and line 254, page 7: “is not convenience indicate two letters, only one letter can go in figure 1 and 3, table 2”.
Thank you for your valuable advice. After our careful consideration about the comments, could we offer our own thoughts on the differential display lettering? If possible, we would like to say that we used pairwise comparisons between groups to analyze the data. If only one letter is used, there may be missing results between this group of data and another group of comparison data. Therefore, please consider carefully and allow us to keep the original copy to ensure that the data are comprehensive. Thank you.

Reviewer 3 Report
The comments are in the manuscript.

Author Response
Dear Reviewer:
Thank you for your letter and the comments about our manuscript entitled “Turn Waste into Treasure: Spent Substrates of Auricularia heimuer Can be Used as the Substrate for Lepista sordida Cultivation” (horticulturae-2595871). These comments are valuable and will be very helpful for us to revise and improve our paper, as well as to help provide the important guiding significance of our research. We studied the comments carefully and made corrections that we hope will be met with your approval. The revised portions are marked in yellow in the paper.
- Line 99 -100, page 3: “g.L-1”and line 153, page 4: “kg.m-2”
Thank you for your valuable advice. We replaced “g.L-1” with “g L-1.” In addition, we replaced “kg.m-2” with “kg m-2.” and “g.bag-1” with “g bag-1”. All of these words have been corrected throughout the manuscript.
- Line 106, page 3: a grinding machine, please write the name of instruments.
Thank you for your valuable advice. We added the appropriate information for the grinding machine as follows: a grinder (Xinzhuohui-FSJ, Zhuohui Machinery Co., Ltd., Zhengzhou, China).
- Line 111, page 3: a crusher, please write the name of instruments.
Thank you for your valuable advice. We added the appropriate information for the crusher as follows: a crusher (M-ZR, Nongxing Machinery and Equipment Factory, Xingyang, China)
- Line 114-116,page 3 : “The contents of carbon (C) and nitrogen (N) were analyzed as described by Dundar et al. [23]. The contents of cellulose, lignin, and hemicellulose were determined as described by Goering et al. [24]”. Please describe these methods!
Thank you for your valuable advice. We described these methods as follows:
The dry matter (DM) of the samples was determined after they had been dried to a constant weight at 65 °C. The contents of total carbon (TC) and total nitrogen (TN) were estimated by the loss of ignition and Kjeldahl methods, respectively. The C/N ratio is the ratio of the TC in the substrate to that of the TN. The contents of ash-free neutral detergent fiber (NDF) [23], ash-free acid detergent fiber (ADF) and lignin [24] were determined as described. The hemicellulose content was calculated as the difference between NDF and ADF and that of cellulose as the difference between ADF and acid detergent lignin (ADL).
- Line 116-118, page 3: “The total C content of the SSA was 42.18%, while that of the total N was 1.89%. The C/N ratio of the SSA was 22:1. The contents of cellulose, lignin, and hemicellulose in the SSA were 16.00%, 14.90% and 39.10%, respectively.” These are results, these not belong here.
Thank you for your valuable advice. We moved them to the results.
3.1. Components of SSA
The TC content of the SSA was 42.18%, while that of the TN was 1.89%. The C/N ratio of the SSA was 22:1. The contents of cellulose, lignin, and hemicellulose in the SSA were 16.00%, 14.90% and 39.10%, respectively.
- Line 130-131, page 3: “The mixture was placed in a trapezoidal pile with two rows of air holes that had a diameter of 5 cm and were spaced at 50 cm intervals.” This is more likely composting process, not fermentation process.
Thank you for your concern about this. the substrate fermentation described in our paper is indeed similar to the composting process. Mushroom substrate fermentation refers to the process of rapid transformation of organic plant matter into a structured, selective, nutrient-rich medium to produce mushrooms, but the processes may be slightly different depending on the type of edible fungi that are cultivated. Due to the different application direction of fermentation products (fertilizer and cultivation substrate), there are similar but also some differences in the specific process of fermentation. Substrate fermentation generally contains two standard stages, including one for the inactivation of pathogens and another for the substrate transformation. It is generally agreed that composting has four phases, including mesophilic, thermophilic, cooling and maturation.
As with compost, substrate fermentation also requires a certain amount of air, air holes are mainly to increase ventilation, avoid anaerobic environment, affect the effect of fermentation, resulting in changes in metabolites.
- Line 280, page 9: “100 g-1” please indicate dry or wet?
Thank you for your valuable advice. We added “dry” to the description of the calculation as follows: g 100 g -1 (dry matter).

Reviewer 4 Report
line 49. Delete "of" after "species"
line 50. Replace "it" by "its cultivation"
line 59. What does "pot" mean? [I have never seen this unit before.]
line 62. Delete "The", starting the sentence with "Successful"
lines 121-125. These lines simply repeat what the reader can observe in Table 1; delete them.
line 197. Replace "p<0.05" by "α=0.05". [The Greek letter alpha denotes a significance level, and that is what is appropriate here. The post hoc LSD test is carried out at a specific significance level.]
line 219. Replace "p<0.05" by "α=0.05"
line 257. Replace "p<0.05" by "α=0.05"
line 283. Replace "p<0.05" by "α=0.05"
line 332. Replace "fungi" by "fungus"
line 346. Replace "of fruiting body" by "of the fruiting body"
line 347. Replace "was no more data" by "were no more data"
Supplementary Table S1. Delete the sentence "The same letter indicates no significant differences." Also, replace "P<0.05" by "α=0.05"
Supplementary Table S2. Replace "P<0.05" by "α=0.05"
Author Response
Dear Reviewer:
Thank you for your letter and the comments about our manuscript entitled “Turn Waste into Treasure: Spent Substrates of Auricularia heimuer Can be Used as the Substrate for Lepista sordida Cultivation” (horticulturae-2595871). These comments are valuable and will be very helpful for us to revise and improve our paper, as well as to help provide the important guiding significance of our research. We studied the comments carefully and made corrections that we hope will be met with your approval. The revised portions are marked in yellow in the paper.
- line 49. Delete "of " after "species"
Thank you for your valuable advice. We deleted “of.”
- line 50. Replace "it" by "its cultivation"
Thank you for your valuable advice. We replaced "it" with "its cultivation."
- line 59. What does "pot" mean? [I have never seen this unit before.]
We apologize for our use of an inappropriate word. In the study by Li, they put 15 kg of compost in plastic trays to cultivate mushrooms. They used the word “pot” in their English abstract. That is why we used this word.
- line 62. Delete "The", starting the sentence with "Successful"
Thank you for your valuable advice. We deleted “The” and started the sentence with "Successful."
- lines 121-125. These lines simply repeat what the reader can observe in Table 1; delete them.
Thank you for your valuable advice. We deleted lines 121-125.
- line 197. Replace "p<0.05" by "α=0.05". [The Greek letter alpha denotes a significance level, and that is what is appropriate here. The post hoc LSD test is carried out at a specific significance level.]
Thank you for your valuable advice. We replaced "p<0.05" with "α=0.05."
- line 219. Replace "p<0.05" by "α=0.05"
Thank you for your valuable advice. We replaced "p<0.05" with "α=0.05."
- line 257. Replace "p<0.05" by "α=0.05"
Thank you for your valuable advice. We replaced "p<0.05" with "α=0.05."
- line 283. Replace "p<0.05" by "α=0.05"
Thank you for your valuable advice. We replaced "p<0.05" with "α=0.05."
- line 332. Replace "fungi" by "fungus"
Thank you for your valuable advice. We replaced "fungi" with "fungus."
- line 346. Replace "of fruiting body" by "of the fruiting body"
Thank you for your valuable advice. W replaced "of fruiting body" with "of the fruiting body."
- line 347. Replace "was no more data" by "were no more data"
Thank you for your valuable advice. We replaced "was no more data" with "were no more data."
- Supplementary Table S1. Delete the sentence "The same letter indicates no significant differences." Also, replace "P<0.05" by "α=0.05"
Thank you for your valuable advice. We deleted the sentence "The same letter indicates no significant differences." and replaced "p<0.05" with "α=0.05."
- Supplementary Table S2. Replace "P<0.05" by "α=0.05"
Thank you for your valuable advice. We replaced "p<0.05" with "α=0.05."

Round 2
Reviewer 3 Report
The paper can be accepted in this form.